

# Dual-frequency radar observations of snowmelt processes on Antarctic perennial sea ice by CFOSCAT and ASCAT

Rui Xu[1], Chaofang Zhao[2,3], Stefanie Arndt[4], Christian Haas[4,5]

[1]College of Atmospheric Sciences, Sun Yat-sen University, Zhuhai, 519082, China
[2]College of Marine Technology, Ocean University of China, Qingdao, 266100, China
[3]Laboratory for Regional Oceanography and Numerical Modeling, Qingdao Marine Science and Technology Center, Qingdao, 266237, China
[4]Alfred-Wegener-Institut Helmholtz-Zentrum für Polar- und Meeresforschung, Bremerhaven, 27515, Germany
[5]Institute for Environmental Physics, University of Bremen, 28359, Germany

*Correspondence to*: Chaofang Zhao (zhaocf@ouc.edu.cn)

**Abstract.** Since 2017, Antarctic sea ice coverage has shown strong reductions, and therefore observations of its surface melting behavior are of utmost importance. Here we study the capability of the Ku-band Chinese-French Oceanography Satellite Scatterometer (CFOSCAT) launched in 2018 to

detect surface melting and compare it with more established observations of the C-band Advanced Scatterometer (ASCAT) in orbit since 2007. Both CFOSCAT and ASCAT observations show increases of radar backscatter of more than 2 dB over perennial ice once the ice surface warms and destructive snow metamorphism commences, defined as pre-melt onset (PMO). Backscatter increases by more than 3 dB once prominent thaw-freeze cycles commence, defined as snowmelt onset (SMO). Scatterometer

data are compared with drifting buoy and ERA5 reanalysis air temperature data to support the interpretation of melt-related snow processes. Between 2019 and 2022, the average CFOSCAT pre-melt and snowmelt onset dates for 12 perennial ice study regions are Nov 9±23 days and Dec 1±22 days and earlier than those of ASCAT on Nov 21±22 days and Dec 11±25 days, respectively. Sensitivity tests show that results slightly depend on chosen backscatter thresholds but little on sea ice concentration. The

derived SMO are in good agreement with previous studies, but the SMO difference between dual-frequency radar observations is smaller than that reported by previous studies due to the sensor differences and different spatiotemporal resolutions. SMO differences between dual-frequency radar observations were also found to be potentially related to regional differences in snow metamorphism. With regard to the long-term changes in SMO, there is strong interannual and regional variabilities in

SMO changes and no clear changes could be detected concurrently with the beginning of Antarctic sea ice decline after 2015. Dual-frequency CFOSCAT and ASCAT observations hold strong promise for better understanding of snowmelt processes on Antarctic sea ice and it is necessary to extend the observation of Antarctic snowmelt based on dual-frequency scatterometers.

## 1 Introduction

After many decades of slight increase, Antarctic sea ice extent has suddenly begun to plummet in 2016 and, after some years of slight recovery, has reached record-low values since 2022 particularly in summer (Turner et al., 2022). However, the causes of these sudden changes and the future fate of Antarctic sea ice are unclear, and different underlying thermodynamic and dynamic atmospheric and oceanic processes



should be considered (Yu et al., 2022; Purich and Doddridge, 2023). In the Arctic, the ongoing reductions
in sea ice extent and thickness are accompanied by earlier melt onset and a longer melt season, as can be
well observed with passive and active satellite microwave data due to the presence and persistence of
wet snow and melt ponds in summer leading to sustained reductions of radar backscatter (Mortin et al.,
2014; Forster et al., 2001). However, Antarctic sea ice does not show strong surface melt due to small
atmospheric energy fluxes (Andreas and Ackley, 1982; Vihma et al., 2009). Most of the melt takes place
at the bottom due to generally large ocean heat fluxes in most parts of the Southern Ocean and Antarctic
marginal seas (Schroeter and Sandery, 2022; Purich and Doddridge, 2023). Instead, the snow cover is
subjected to diurnal or multi-day thaw-freeze cycles with strong snow metamorphism and the formation
of ice layers or superimposed ice (Haas et al, 2001; Haas et al., 2008; Arndt et al., 2021). On perennial
ice, i.e. the ice surviving spring and summer, this leads to sustained decreases in microwave emissivity
(Willmes et al., 2009; Arndt et al., 2016) and increases in radar backscatter which can be used to observe
the timing of snow melt onset and subsequent refreezing (Haas, 2001; Willmes et al., 2006; Arndt and
Haas, 2019).

Radar scatterometers play an important role in investigating melt processes on Antarctic sea ice due to
their frequent revisit of the entire sea ice region. Strong snow metamorphism results in coarse-grained
and salt-free summer snow, increasing radar volume- and surface-scattering during the spring-summer
transition defined as the period from Oct 1 to Jan 31 of the following year (Haas, 2001). When the ice
melts before strong snow metamorphism due to thaw-freeze cycles commences, e.g. in the seasonal sea
ice zone, backscatter varies strongly once surface flooding with sea water of the thinning ice begins (e.g.
Massom et al., 2001), and clear melt events cannot be easily detected (Drinkwater and Lytle, 1997;
Drinkwater and Liu, 2000; Arndt and Haas, 2019). Therefore, this study focuses on the snowmelt onset
in the perennial sea ice region only.

Using the increasing backscatter behavior when sporadic snowmelt begins, Arndt and Haas (2019)
retrieved the pre-melt and snowmelt onsets from Ku-band and C-band scatterometer observations of the
Antarctic perennial ice from 1992 to 2014. They showed the backscatter seasonal amplitudes had
latitudinal variations, with Ku-band backscatter responding more strongly to snowmelt than C-band
backscatter. Comparison of snowmelt onset between Ku-band and C-band sensors showed Ku-band
QuikSCAT observations yielded earlier pre-melt and snowmelt onsets than C-band ERS and ASCAT
observations. Arndt and Haas (2019) also proposed a conceptual model which suggested that the Ku-
band signal first detected the initial snow property changes in the inner snow column. The C-band signal
can only sense the snowmelt when the warming and refreezing of snow reaches the deeper layers. This
demonstrates that dual-frequency microwave satellite observations can help resolve the vertical snow
column melting process of thick snow on the Antarctic perennial sea ice from space and opens new ways
to study the energy and mass budget of snow on sea ice in the Southern Ocean.

The Ku-band QuikSCAT scatterometer has significantly contributed to snowmelt studies in the polar
region (Arndt and Haas, 2019; Sturdivant et al., 2018; Wang et al., 2018). However, the end of its life



span in 2009 meant that observations were not possible any more. Fortunately, in October 2018 the
Chinese-French Oceanography Satellite Scatterometer (CFOSCAT) was successfully launched,
providing valid Ku-band measurements with 12.5 km spatial resolution at a frequency of 13.4 GHz that
coincide with observations of the C-band ASCAT from 2019 to 2022.

In this study we analyze the capability of CFOSCAT to detect melt onset timing, and compare its
performance to observations from ASCAT, closely following, extending, and updating the work and
methods of Arndt and Haas (2019). However, in addition we carry out sensitivity analyses to justify
thresholds chosen for required backscatter increases and for the minimum ice concentration for which
our results are still valid. We significantly extend the work of Arndt and Haas (2019) by studying the
relationship between melt detection and snow melt processes more carefully by comparing results with
air temperature time series from the ERA5 reanalysis. Finally, we use data from a drifting snow buoy to
compare changes of backscatter along the drift trajectory with changes of air temperature and snow depth
observed by the buoy. Our regional analysis is based on 12 study sites on perennial ice (Fig. 1) identical
to those by Arndt and Haas (2019).

## 2 Data and methods

### 2.1 Data set

Scatterometer data used here include Ku-band CFOSCAT data and C-band ASCAT data. Despite the
Chinese-French Oceanography Satellite having a designed lifetime of 3 years, it has exceeded this
timeframe and remines in operation. However, since December 2022, the antenna of its scatterometer
payload CFOSCAT has stopped spinning and has remained stalled in a fixed azimuth angle, unable to
provide valid data (Mou et al., 2023). The CFOSCAT operates at a frequency in Ku-band (13.4 GHz)
with medium incidence angles (28°-51°, see Liu et al., 2020). It applies two fan-shaped beams to scan
the earth, one of which is a vertically polarized (v-pol) fan beam, and the other is horizontally polarized
(h-pol).

The ASCAT on MetOp operates at a frequency in C-band (5.255 GHz) and provides vertical polarization
data. Here we used gridded enhanced ASCAT data from the NASA Scatterometer Climate Record
Pathfinder (SCP) project provided by Bigham Young University (BYU) that are 2-day averages with a
resolution of 4.45 km×4.45 km (Long et al., 1993). ASCAT data have been available from SCP since
2007 and are currently updated to 19 November 2022, all of which were used to complete this study.


As fan-beam scatterometers, ASCAT and CFOSCAT obtain backscatter coefficients with various
incidence angles. This requires incidence angle corrections to eliminate the effect of incidence angles on
backscatter coefficients. BYU uses a scatterometer image reconstruction with filter algorithm (SIRF;
Long et al., 1993) to correct the ASCAT data by bringing backscattering coefficients to a reference
incidence angle of 40°. In order to achieve the incidence angle correction of the CFOSCAT
measurements, we used a robust least squares fitting method (Xu et al., 2022) based on the CFOSCAT



Level 2A data. This method is based on the linear relationship between the backscatter coefficient (in dB) and the incidence angle (Remund and Long, 1999) as follows,

$$\sigma_o^0(\theta) = A + B(\theta - 40°) , \tag{1}$$

where $A$ is the observed backscatter value ($\sigma_o^0$) at 40° incidence, i.e., the backscatter after incidence angle correction, and $B$ describes the dependence of $\sigma_o^0$ on $\theta$. First, we projected the 12.5 km resolution CFOSCAT L2A data onto the same polar stereographic grid as the ASCAT data. Then the least squares fitting was performed for polar grid cells with more than 5 observations. To ensure the same spatial and temporal resolution between ASCAT and CFOSCAT, we downsampled the ASCAT backscatter data
from 4.45 km to 12.5 km resolution by averaging the data within each 12.5 km grid cell, and averaged the CFOSCAT data over two days. Note that we used v-pol CFOSCAT backscatter to be consistent with the polarization of the ASCAT data, as also done by Arndt and Haas (2019).

To avoid the contamination of our results by wind-roughened water and regions with low ice
concentration, we used the sea ice concentration (SIC) data derived from the Advanced Microwave Scanning Radiometer (AMSR) series to filter out scatterometer data at low sea ice concentrations. We only considered scatterometer data if the ice concentration was more than 70%, a threshold chosen as a robust value from our sensitivity study below. We used the daily AMSR SIC data product developed by Spreen et al. (2008), which has a spatial resolution of 6.25 km.


In order to help interpret our melt detection results we used hourly-air temperature data from ERA5 reanalysis (Hersbach et al., 2023). These data are derived from observations and data assimilation in weather models, and have been widely utilized and evaluated for the Antarctic region (Bozkurt et al., 2020; King et al., 2022; Zheng et al., 2022). Here we used the 2-m ERA5 air temperature with a spatial
resolution of 0.25° × 0.25°.

Finally, we evaluated our results with air temperature and snow depth data from a snow buoy. Snow buoys are autonomous, drifting platforms deployed on individual ice floes (Nicolaus et al., 2021). Unfortunately, during the CFOSCAT observation period from 2019 to 2022 only data from one buoy are
available, namely 2021S114 (hereafter S114). It was deployed in the Weddell Sea on Feb 19, 2021, and provided continuous measurements through the 2021/2022 melt season. Its drift track between October 1, 2021 and January 31, 2022 is shown in Fig. 1.



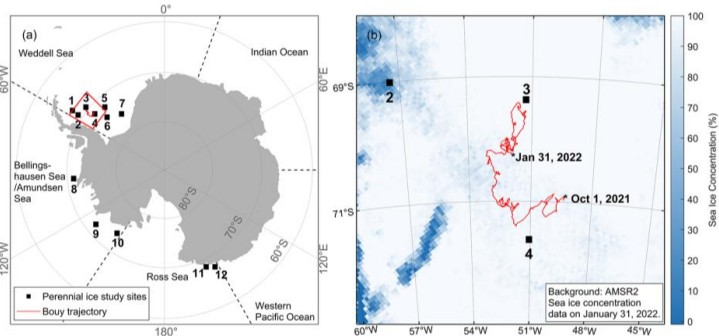


**Figure 1.** (a) Locations of 12 study sites (black squares) in the Antarctic perennial sea-ice zone and the drift trajectory of buoy S114 (red line). We define abbreviations for different sectors for subsequent use: WS is Weddell Sea, BS/AS is Bellingshausen Sea and Amundsen Sea, RS is Ross Sea, WPO is Western Pacific Ocean, and IO is Indian Ocean. (b) Drift trajectory of buoy S114 between October 1, 2021 and

January 31, 2022 (red line). Background shows sea ice concentration on January 31, 2022. Locations 2, 3, and 4 are the same on both maps.

## 2.2 Snow melt retrieval

In the Antarctic, snowmelt is not strong enough to form widespread melt pond coverage due to relatively low net surface heat gain (Vihma et al., 2009). Therefore, methods used in the Arctic that take advantage

of strong backscatter decreases due to wet snow and ice are not applicable in the Antarctic. In contrast, on Antarctic sea ice, most snow survives the summer but undergoes strong diurnal thaw-freeze cycles leading to highly metamorphous snow with high backscatter (Massom et al., 2001; Haas, 2001; Willmes et al., 2009; Arndt and Haas, 2019).

This is illustrated in Fig. 2, an example of annual time series of CFOSCAT and ASCAT radar backscatter. It can be seen that backscatter generally decreases during winter, with shorter-term variations of less than 2 dB. However, during November, larger backscatter increases of more than 2 dB can be seen. We define this phase as pre-melt onset (PMO). It coincides with the spring warming, when the snow and upper layers of ice begin to warm due to higher air temperatures and increasing solar radiation. If snow

temperatures get closer to the melting point, liquid water appears between grains in the snow, initially in the pendular regime, i.e. without percolating (Colbeck, 1997). Under the influence of diurnal temperature and radiation cycles, it leads to the formation of increasingly large, round crystal clusters, which enhance the volume scattering of snow and thus result in radar backscatter rise (Nandan et al., 2017; Arndt and Haas, 2019). In addition, also the upper ice warms during this period, and becomes more porous

depending on its salinity, contributing to increased backscatter as well (Yackel et al., 2007).





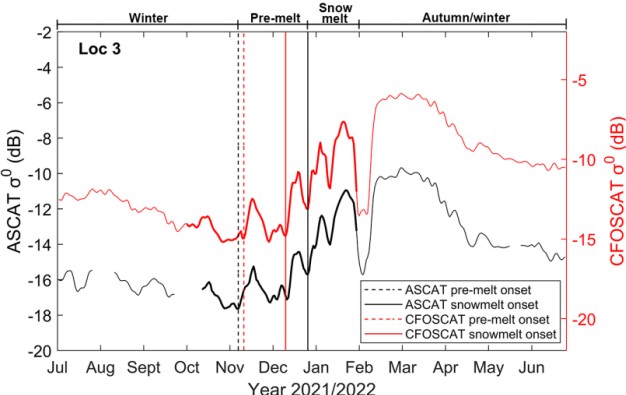

**Figure 2.** Annual time series of CFOSCAT ($\sigma^0_{CFOSCAT}$, red) and ASCAT radar backscatter ($\sigma^0_{ASCAT}$, black) in 2020/2021 for one pixel at Location 3 in the perennial sea ice zone (see Fig. 1). Vertical lines illustrate snow pre-melt onset (stippled) and snowmelt onset (solid) derived from the backscatter time series. Bolded section denotes the period of the spring-summer transition (Oct 1 to Jan 31).

In December and January, larger variations of backscatter can be seen, with more persistent increases over 3 dB alternating with short intermittent decreases. We define the first rise of more than 3 dB as snowmelt onset (SMO). It coincides with the onset of more extensive liquid water within the snow and strong diurnal thaw-freeze cycles. These lead to strong snow metamorphism and the occasional transformation of the snow to the funicular regime (Colbeck, 1997), when meltwater percolates downwards and refreezes within the snow or at the colder snow-ice interface, depending on the temperature profile within the snow. When the melt is intense so that meltwater percolates to the snow-ice interface and refreezes there, it forms fresh superimposed ice on top of the original sea ice (Jeffries et al., 1994; Haas et al., 2001; Kawamura et al., 2017). Metamorphic snow, ice layers, and superimposed ice all contribute to strong backscatter rises. Short periods of several days of wetter snow with little refreezing cause the observed intermittent backscatter drops.

This characteristic temporal snow and backscatter behavior can be seen in most regions of perennial ice, and was used by Arndt and Haas (2019) to derive pre-melt and snowmelt onset from ERS, QuikSCAT, and ASCAT data. Here we follow the same approach by first extracting the annual backscatter time series of ASCAT and CFOSCAT for each study grid cell from the beginning of July to the end of June of the following year, as shown in Fig. 2. In contrast to Arndt and Haas (2019) who resampled ASCAT and QuikSCAT backscatter time series to time intervals of 6 days to be consistent with the temporal resolution of the ERS data, in this study we did not resample the ASCAT data but used their original 2-day resolution. We averaged the daily CFOSCAT data over two days to align with the temporal resolution of ASCAT (Section 2.1). Therefore, our study can potentially capture shorter periods of snow property changes. As done by Arndt and Haas (2019), we smoothed each time series by a moving three-point averaging window to reduce noise, such as from wind or spurious warm intrusions.



Based on the backscatter time series, we calculated the differences between the local backscatter maxima and preceding local backscatter minima during the spring-summer transition. The first instance after Oct 1, when the difference is larger than a pre-melt onset threshold, is regarded as the pre-melt onset date (vertical stippled lines in Fig. 2). The first instance after the pre-melt onset date when the difference is larger than a snowmelt onset threshold is regarded as the snowmelt onset date (vertical solid lines in Fig. 2). The study of Arndt and Haas (2019) found that 2dB is a suitable threshold for PMO, and 3dB for SMO, respectively. Since these two thresholds were derived from a long-term backscatter compilation including different scatterometers and climatic conditions, we used the same threshold for ASCAT and CFOSCAT data in this paper. However, note that we carried out an additional sensitivity study which justifies these choices (Section 3.1).

For better comparison with previous studies, we focused on the same 12 perennial ice study sites containing 3 x 3 pixels selected by Arndt and Haas (2019) and Haas (2001) from different Antarctic sectors (Black squares in Fig. 1a). We retrieved the pre-melt and snowmelt onsets for each of those 9 pixels and then averaged their results to reduce noise and to increase the spatial representativity.

## 3 Results

### 3.1 Sensitivity of PMO and SMO dates to different backscatter thresholds

Before using thresholds of 2dB for PMO and 3dB for SMO, it is necessary to test their effectiveness. Therefore, we carried out a sensitivity study by successively increasing and decreasing the thresholds by 0.1 dB up to a difference of +/- 1 dB, and retrieving the PMO and SMO over the 12 study sites with these backscatter thresholds. Fig. 3 shows the changes in PMO and SMO in different perennial ice regions during the period when CFOSCAT and ASCAT data overlap, i.e., from 2019/2020 to 2021/2022. It can be well seen that both PMO and SMO become progressively later as the threshold increases. The larger the threshold, the smaller the change, especially when the threshold is greater than 2 dB for PMO and 3 dB for SMO. The blue line in Fig. 3 indicates the retrieval rate, defined as the ratio of pixels within the 12 9-pixel study sites where PMO or SMO can be retrieved to all 108 pixels. Generally, the retrieval rate decreases as the threshold increases. When the threshold is greater than 2 dB, the PMO retrieval rate of ASCAT drops below 0.99, while the PMO retrieval rate of CFOSCAT stays above 0.99. The SMO retrieval rate of ASCAT is also lower than that of CFOSCAT, especially when the threshold is greater than 3 dB, the SMO retrieval rate of ASCAT drops below 0.88.

Based on the threshold sensitivity test, we conclude that 2 dB and 3 dB are appropriate thresholds to avoid underestimating the PMO and SMO, while the choice of thresholds may still affect results by up to a week or so. On the other hand, thresholds larger than 2dB and 3dB might be more difficult to be satisfied frequently which would reduce the retrieval rate. We also found that CFOSCAT had a higher retrieval rate than ASCAT.





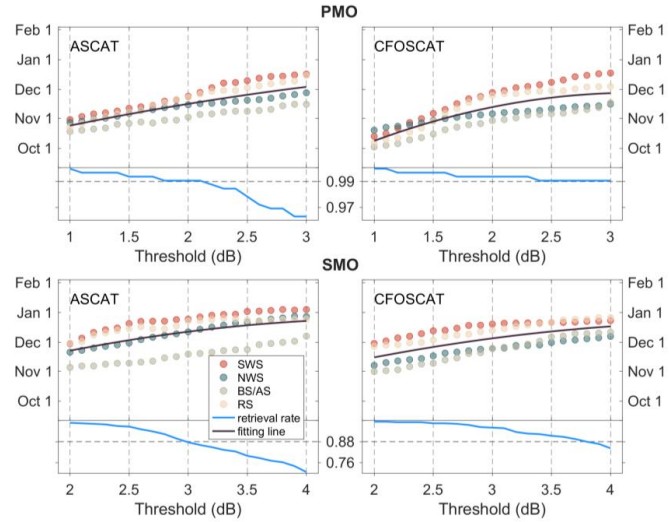

**Figure 3.** Variations of retrieved PMO and SMO dates with the change of retrieval thresholds for different regions. The black line represents the fitted curves for PMO and SMO and the blue line indicates the retrieval rate defined as the fraction of pixels within the 12 study sites where PMO or SMO can be retrieved. SWS: Southeastern Weddell Sea. NWS: Northwestern Weddell Sea. The boundary between SWS and NWS is defined as the longitude of Location 4, i.e., 50.1°W (see Fig. 1). For other full names, see the title of Fig. 1.

### 3.2 Pre-melt and snowmelt onsets retrieved by CFOSCAT and ASCAT

With the thresholds of 2 dB for PMO and 3 dB for SMO now justified, we continue to compare the PMO and SMO detections from CFOSCAT and ASCAT. Figure 4 shows the daily backscatter time series as well as the retrieved pre-melt and snowmelt onsets of CFOSCAT and ASCAT for our 12 study regions. The Ku-band backscatter shows higher values than the C-band backscatter during winter, in agreement with results of Mortin et al. (2014) who compared QuikSCAT and ASCAT in the Arctic. This points to the importance of snow properties and the different penetration depths of Ku- and C-band radar on radar backscatter (e.g., Onstott and Shuchman, 2004). During winter, both CFOSCAT and ASCAT backscatter decreases, probably due to increasing new snow accumulation and the opening of leads with the young first-year ice formation within the divergent ice pack. In the melt season, the backscattering coefficients increased significantly during the spring-summer transition, and more strongly with CFOSCAT. This can be seen in Fig. 5 which summarizes the mean amplitudes between the seasonal maxima and minima during the spring-summer transition for CFOSCAT and ASCAT over period of 2019/2020 to 2021/2022. The average increase in CFOSCAT was 8.0 dB, larger than that of ASCAT with 5.9 dB. It is consistent with the findings of Arndt and Haas (2019), in which the seasonal backscatter amplitude of QuikSCAT is 8.03 dB, and that of ASCAT is 5.1 dB, respectively. It suggests that Ku-band scatterometers are more sensitive to the snowmelt process on Antarctic perennial ice than C-band scatterometers. This also results in the higher retrieval rates of CFOSCAT than ASCAT shown in Fig. 3.



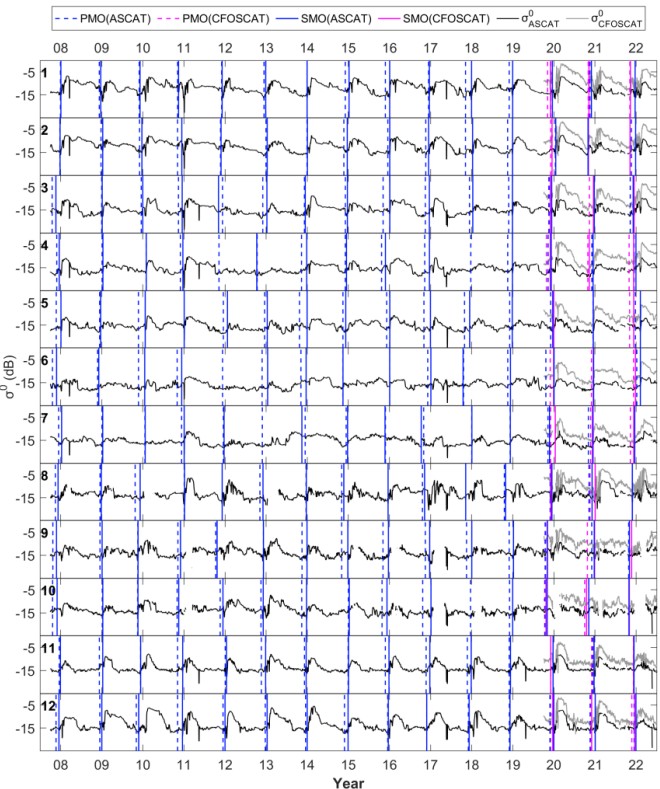

**Figure 4.** Time series of ASCAT (black lines) and CFOSCAT backscatter (gray lines) averaged over the respective nine pixels at the 12 study locations with the retrieved pre-melt (PMO; stippled lines) and snowmelt onset dates (SMO; solid lines) shown by vertical lines.

The latitudinal distribution of the seasonal amplitudes in the 12 study regions can also be observed in Fig. 5. In the Weddell Sea region, ASCAT and CFOSCAT seasonal amplitudes are relatively low at high latitudes (Locations 5, 6, and 7) but higher at low latitudes (Locations 1 to 4). In the Bellingshausen Sea/Amundsen Sea region and Ross Sea region, it is also quite evident that the backscatter amplitude decreases with increasing latitude, however sites are also separated much zonally. This phenomenon suggests that changes in backscatter amplitudes relate to latitude variations owing to cooler climate conditions farther south.



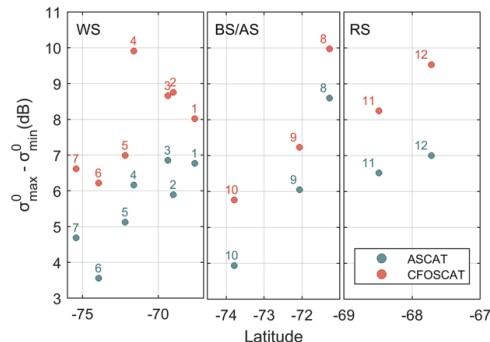

**Figure 5.** Differences between seasonal maxima and minima of backscatter during the spring-summer transition period over 2019/2020 to 2021/2022 for our 12 study sites shown by latitude. See Fig. 1 for locations and region acronyms.

Also the timing of melt onsets varies with latitude and region, with both ASCAT and CFOSCAT showing later snowmelt onset in the Southeastern Weddell Sea than in the Northwestern Weddell Sea. Fig. 6 summarizes the mean pre-melt and snowmelt onset dates derived from CFOSCAT and ASCAT. For the period from 2019/2020 to 2021/22, the average SMO of CFOSCAT is significantly earlier than that of ASCAT for all sub-regions, with the exception of the BS/AS region, where their snowmelt onsets are comparable. This could be due to the fact that the snow in the BS/AS region melts very rapidly when it is already close to the melting temperature and warms quickly throughout the entire snow column. Then all wavelengths would respond at approximately the same time (Arndt and Haas, 2019). On average, CFOSCAT detects snowmelt onset on Dec 1, i.e. ten days earlier than ASCAT (Dec 11). This is in agreement with the observations and conceptual scattering model of Arndt & Haas (2019), although they found a larger average difference of 18 days between Ku- and C-band detected snowmelt onset. As for the pre-melt onset, CFOSCAT detected an average of Nov 9, 12 days earlier than ASCAT (Nov 21), with the difference between the C- and Ku-band sensors much smaller than the 20 days reported by Arndt & Haas (2019). The largest difference (19 days) between CFOSCAT and ASCAT results occurs in the NWS region, where the backscatter amplitude varies significantly during the melt (Fig. 4). We also find that both the mean PMO and SMO of ASCAT from the 2019/2020 to 2021/22 across all regions are earlier than those from the 2007/08 to 2021/22, which may be important for studying the large Antarctic sea ice variability in recent years.





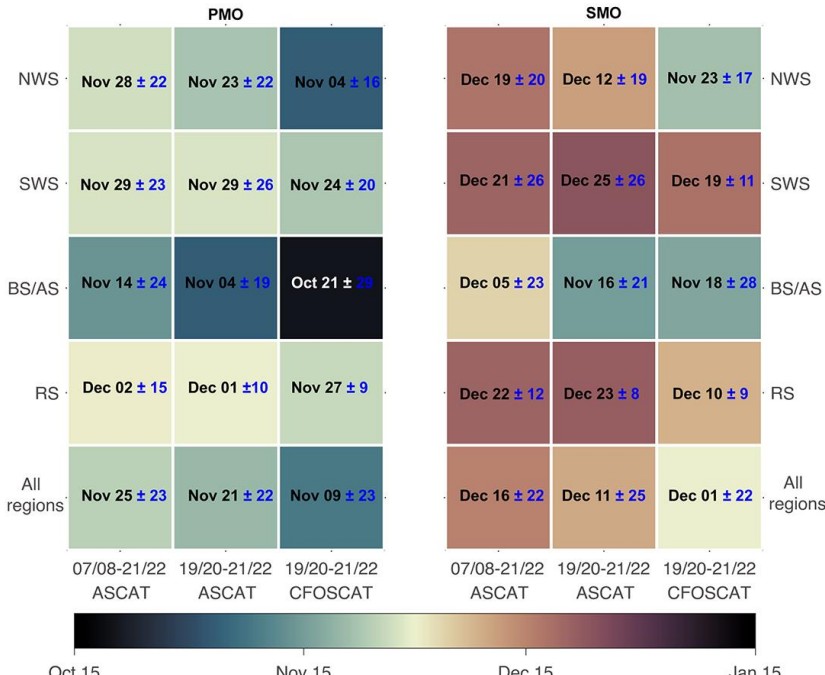

**Figure 6.** Mean values of pre-melt and snowmelt onset dates of ASCAT and CFOSCAT for 12 study regions (mean ± 1 standard deviation) and different time periods. The left panel shows PMO derived from ASCAT for the period 2007/08 to 2021/22, and from ASCAT and CFOSCAT for the period 2019/21 to 2021/22. The right panel shows the statistics of SMO for the same time periods. Colors indicate dates.

### 3.3 Relationship between scatterometer-derived melt onset timing and the air temperature

Figure 7 shows time series of 2-day running mean daily maximum ERA5 2-m air temperatures from July 2007 to June 2022 for the 12 study regions, along with scatterometer-derived pre-melt and snowmelt onset dates. When October comes, air temperatures begin to rise noticeably. When pre-melt onset is detected, air temperatures have generally risen above -5 ℃.

Looking for single days with high air temperatures may not be sufficient for the interpretation of melt signals, because snow may not thaw or melt when high air temperatures are not sustained. Therefore, in order to find more robust relationships between scatterometer-derived PMO and SMO dates and air temperatures, we also compare PMO and SMO dates with the date when air temperatures have been equal to or higher than 0 ℃ for at least 3 consecutive days ($DATE_{0-3d}$; Fig. 8). The figure also compares PMO with $DATE_{-5}$ (the date when air temperatures first rise above -5 ℃) and $DATE_0$ (the date when air temperatures first rise above 0 ℃), and SMO with $DATE_0$. In order to better compare the results between the C- and Ku-band sensors, only the results from 2019/20 to 2021/22 are shown here.



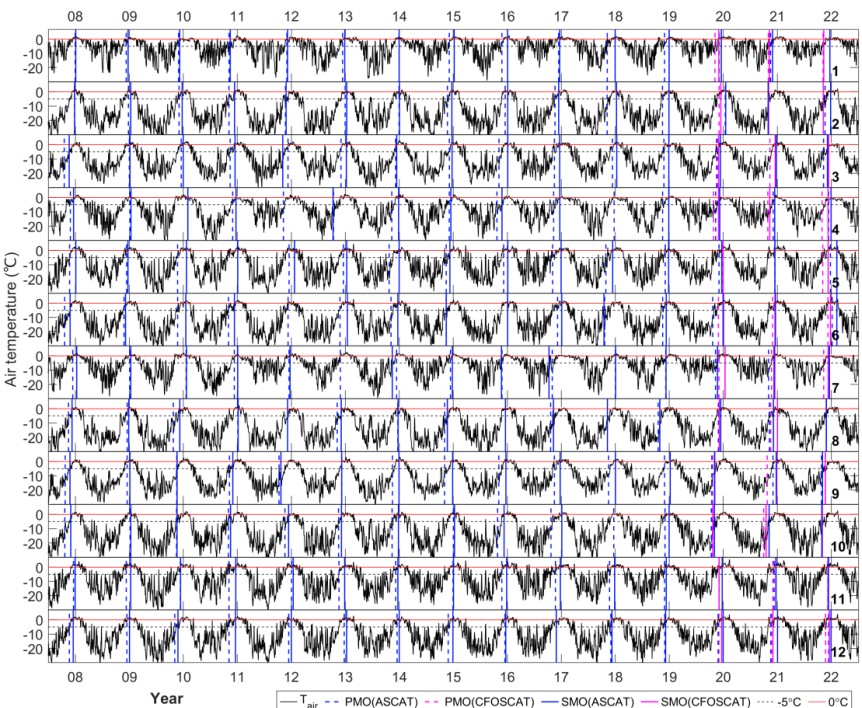

**Figure 7.** Time series of 2-day average maximum ERA5 2-m air temperatures at the 12 study locations
with pre-melt and snowmelt onset dates marked by colored vertical lines.

There are 27 cases of CFOSCAT and 28 cases of ASCAT (Fig. 8a) where the pre-melt onset occurs after
air temperatures first rise above -5 °C, indicating that pre-melt rarely occurs when the air temperature is
still below -5 °C. The PMO of both sensors are closer to the date when temperatures first rise above 0 °C,
with 13 and 16 cases falling within ±10 days of $DATE_0$ for CFOSCAT and ASCAT, respectively (Fig.
8b). In contrast, the PMO dates are mostly earlier than $DATE_{0-3d}$ (Fig. 8c). This suggests that pre-melt
onset derived in this study is sensitive to initial warming and occurs before sustained warming sets in.

Figure 8 (d-e) shows that SMO cases are usually later than $DATE_0$, and most fall within ±20 days of
$DATE_{0-3d}$. It indicates that the SMO retrieved here are more likely to represent a sustained warming
processes. Overall, although the PMO is closer to $DATE_0$ and the SMO is closer to $DATE_{0-3d}$, the
difference between PMO and $DATE_0$ or the difference between SMO and $DATE_{0-3d}$ is sometimes very
large and can exceed 1 month. On the one hand, it may be because $DATE_0$ or $DATE_{0-3d}$ in some cases
correspond to the air temperature rise caused by a short-lived warm intrusion event, and cannot represent
the beginning of a real seasonal pre-melt or snowmelt process. On the other hand, in regions where
melting is also affected by ocean circulation or sea ice drift in addition to air temperature (i.e. Amundsen




Sea region), the PMO or SMO retrieved by scatterometers may have a weakened connection with air temperature, resulting in large differences between PMO and $DATE_0$ or between SMO and $DATE_{0-3d}$.

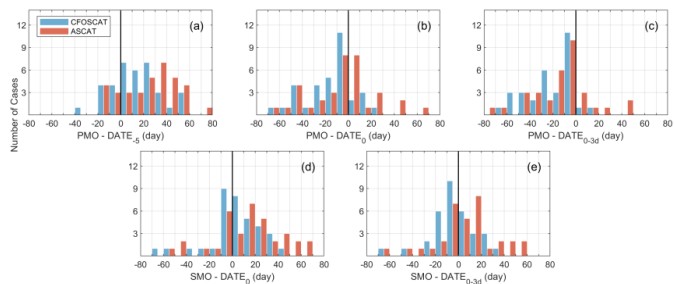

**Figure 8.** Histograms of differences between scatterometer-derived PMO and SMO and the date when air temperatures first rise above -5 °C ($DATE_{-5}$), date when air temperatures first rise above 0°C ($DATE_0$), date when air temperatures have been equal to or higher than 0°C for at least 3 consecutive days ($DATE_{0-3d}$). (a): PMO-$DATE_{-5}$; (b): PMO-$DATE_0$; (c): PMO-$DATE_{0-3d}$; (d): SMO-$DATE_0$; (e): SMO-$DATE_{0-3d}$. Blue represents the results of CFOSCAT, and red represents the results of ASCAT.

**3.4 Comparison with snow buoy data in 2021/22**

Figure 9 shows the seasonal evolution of snow depth and air temperature from buoy S114, ERA5 air temperature, AMSR SIC, and backscatter during the spring-summer transition in 2021/2022 along the buoy drift track. It also shows the pre-melt and snowmelt onset dates retrieved from the backscatter time series along the buoy track using the method described in Section 2.2.

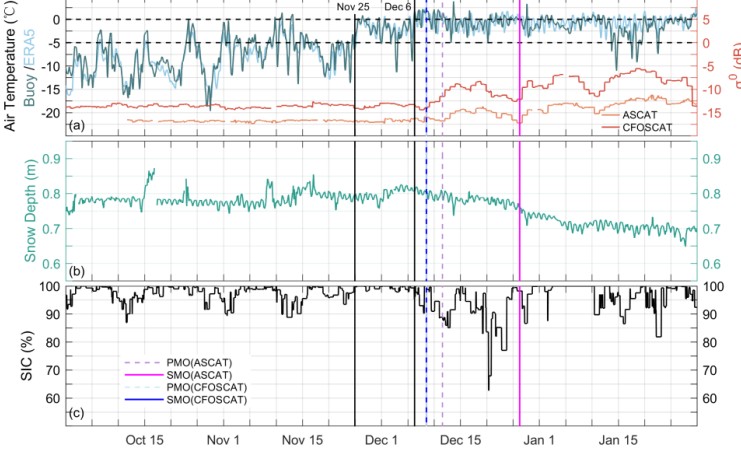

**Figure 9.** Time series of (a) air temperatures measured by snow buoy S114 and by ERA5, backscatter of CFOSCAT and ASCAT, (b) snow depth measured by S114, and (c) sea ice concentration along the drift trajectory of S114 during the spring-summer transition in 2021/22. Snow buoy data and ERA5 air temperatures shown here are hourly data. The PMO and SMO of CFOSCAT occurred on the same day, therefore the two lines cannot be distinguished well.



From Oct 1 to CFOSCAT pre-melt onset, CFOSCAT backscatter remains around -13 dB, while ASCAT maintains a lower backscatter coefficient (-17 dB) than CFOSCAT, in agreement with results shown in Fig. 4. The air temperature gradually increased starting Nov 1, remaining mostly above -5°C after Nov 25. On Dec 6, the air temperature reached and exceeded 0°C. After keeping the temperature above -1.8°C
for three days, CFOSCAT PMO and SMO were detected (Dec 9) due to the significant backscatter change caused by snowmelt. This was followed by a gradual decrease in temperature, in which case the resulting refreezing kept the backscatter of CFOSCAT and ASCAT at high values of -8 and -14 dB, respectively. On Dec 11, ASCAT detected the pre-melt onset. On Dec 17, the air temperature began to rise, increasing the liquid water content in the snow. As the increase in liquid water content causes the
microwave signal to attenuate, the backscatter of both sensors decreased. On Dec 25, air temperatures dropped again with further refreezing that might have led to the formation of ice lenses in the deeper snow layers or superimposed ice on the ice surface. This resulted in an increase of more than 3 dB in backscatter at ASCAT (Dec 26), which allowed ASCAT to detect the onset of snowmelt. It also should be noticed that before Dec 26 backscatter first decreased due to thawing caused by the warmer air
temperature and then increased due to refreezing caused by the cooler air temperature, characterizing the typical microwave backscatter changes during thaw-freeze cycles. Until Jan 31, several other thaw-freeze cycles can also be observed through backscatter variations (e.g., from Jan 8 to Jan 21).

When the SMO of CFOSCAT and ASCAT were detected, the buoy S114 was located at (-70.2° S, -52.3° W) and (-69.6° S, -51.3° W), respectively, both somewhere between Locations 3 and 4, closer to Location
380   W) and (-69.6° S, -51.3° W), respectively, both somewhere between Locations 3 and 4, closer to Location 3 (Fig. 1b). However, the snowmelt onsets obtained along the buoy track were closer to the average of Location 4 (SMO of Dec 9 and Dec 26 for CFOSCAT and ASCAT respectively, Fig. 4) instead of Location 3 (SMO of Nov 16 and Dec 2 of CFOSCAT and ASCAT respectively, Fig. 4) for both sensors. This was due to the fact that when the buoy drifted near Location 4, the local temperature was still very
low, making it difficult to cause snowmelt. The buoy then continued to drift past Location 4 until approaching Location 3, when the backscatter increased significantly and the SMO was detected. At this time, the snowmelt at Location 3 has already started, but at Location 4 it has just started. This phenomenon indicates that Lagrangian and Eulerian measurements may perform differently in retrieval of snowmelt.

Before the pre-melt period, snow depth remained nearly constant or increased slightly, but began to decrease around Dec 7, coinciding with CFOSCAT detected pre-melt. Snow depth continued to decrease until around Jan 6 when it stabilized at around 0.7 m even though air temperatures remained around 0°C for some time afterwards. Figure 9 also shows that SIC remained larger than 80-90% for most of the
time. Larger drops throughout December coincide with periods of higher air temperatures around or above 0°C with the detected pre-melt and snow melt onset events. SIC tends to increase when air temperatures return to below 0°C. While we cannot exclude the fact that ice concentrations may indeed have reduced and may have affected our melt retrieval results, the SIC behavior shown in Fig. 9 may also be due to the impact of snow properties on SIC retrievals (e.g. Spreen et al., 2008).



## 4 Discussion

### 4.1 Limitations and Uncertainties

The melt detection approach in this study is Eulerian, i.e., the influence of ice drift is not considered and therefore it is difficult to strictly distinguish between spatial and temporal variability. The ice drift can redistribute the location of the perennial ice and the snow zone with different properties, affecting the backscatter characteristics of the snow surface, which may in turn increase the uncertainty of snowmelt onset date detection under the Eulerian framework. In order to reduce the small-scale variability caused by small-scale ice drift and ensure stable results, the snowmelt onset date of each study site was computed by averaging the snowmelt onset dates of 9 pixels or 37.5 km x 37.5 km in this paper. With an average ice drift speed of 0.1 m/s, this corresponds to a residence time of approximately four to five days in each study site. However, this still cannot completely eliminate the impact of ice drift. For example, we found in Section 3.4 that snowmelt onset date along the buoy trajectory differs from that at the fixed location near the buoy. A more accurate and quantitative comparison of results based on Lagrangian and Eulerian frameworks requires knowledge of the physical characteristics of each ice field. However, given that no more other snow buoy data or reliable satellite sea ice drift data are available within the temporal and spatial frame of our study, and that this is beyond the scope of our study, no further relevant analysis is conducted in this paper.

Our melt detection results may be affected by the presence of open water whose backscatter can vary widely depending on wind roughening and ice cover. When there is no wind or low wind, open water may reduce the magnitude of the backscatter increase during the melting process, resulting in the defined pre-melt and snowmelt onsets criteria being fulfilled slightly later. In this study, although we filtered out sea ice with a sea ice concentration below 70%, the presence of open water may still have an impact on our retrieval results. Figure 10 shows the sea ice concentration distributions of different regions from 2007 to 2021. It shows that the SIC in the NWS and SWS regions is mostly higher than 70% during the spring-summer transition (99% in NWS region and 98% in SWS region), indicating that our SIC threshold of 70% is justified (Fig. 10). In the BS/AS region, SIC varies significantly during spring-summer transition and can be as low as less than 50% in about 5% of cases. In the RS region, SIC may drop below 70% during the spring-summer transition, but most remain above 70% (95%). Like Arndt et al. (2016), we further analyzed the retrieved pre-melt and snowmelt onset dates over the 12 study regions under five sea ice concentration thresholds (50%, 60%, 70%, 80%, and 90%) as summarized in Fig. 11. Average snowmelt onset dates over all regions of CFOSCAT have no change at different ice concentrations, while the average pre-melt onset has changed by 2 days. For ASCAT, average pre-melt and snowmelt onsets over all regions have changed by 1 day and 2 days respectively from 2019/20 to 2021/22. Fig. 11 shows that BS/AS region is more susceptible to changes in SIC thresholds. In this region, the CFOSCAT SMO varied by 3 days, and the ASCAT SMO changed by 4 days from 2019 to 2022. This is due to the highly variable sea ice concentration in this region as shown in Fig. 10. In the RS region, when the SIC threshold is greater than 70%, the melt onset dates vary significantly. Overall, both average pre-melt and snowmelt onset over all regions vary within 2 days under different sea ice concentrations, indicating that sea ice concentration has a weak impact on our results. Moreover, 70% is





a suitable SIC threshold that will not filter out too many valid data in the WS and RS regions, nor will it cause the retrieval in the BS/AS region to be affected by the presence of too much water.

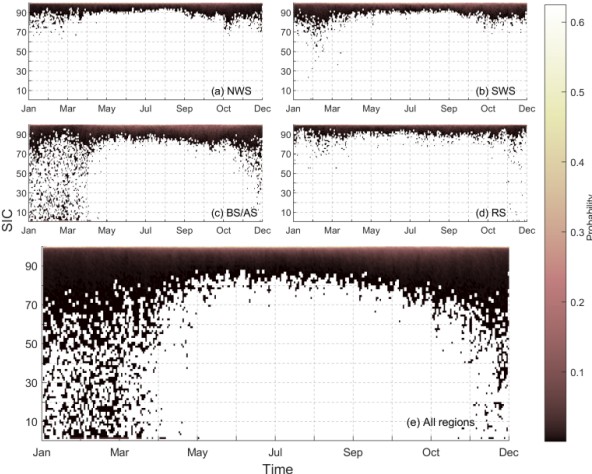

**Figure 10.** Distribution of SIC in different seasons and regions from 2007 to 2021. Colors represent distribution probabilities.

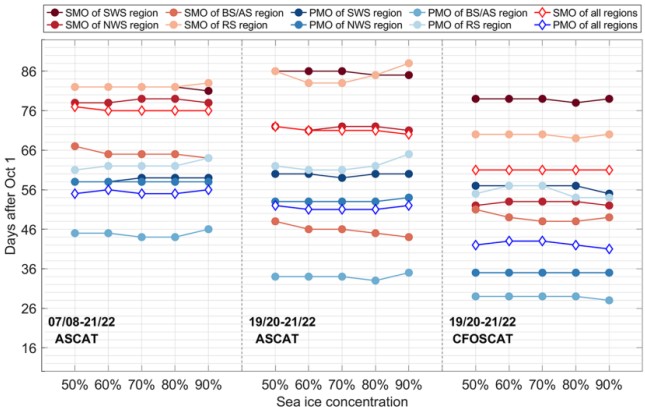


**Figure 11.** Pre-melt and snowmelt onsets for the 12 study sites under different Sea ice concentrations. The values shown are days after Oct 1.

### 4.2 Comparison with the previous study

We used the same thresholds as in Arndt and Haas (2019) to determine pre-melt and snowmelt onset

dates, although Arndt and Haas (2019) derived the results from 6-day average backscatter data time series, whereas we used 2-day average backscatter time series. And Arndt and Haas (2019) resampled the ASCAT data to a 25 km×25 km polar stereographic grid while we resampled the ASCAT data to a 12.5 km×12.5 km polar stereographic grid. Despite differences in temporal and spatial resolution, both studies revealed that Ku-band scatterometers detected earlier snowmelt onsets than C-band scatterometers (Fig.

6 in this study; Fig. 10 in Arndt and Haas (2019)). The reason for this has been demonstrated in the



conceptual model developed by Arndt and Haas (2019), whereby Ku-band scatterometers with shorter penetration depths detect snow metamorphism inside the snowpack earlier than C-band scatterometers with larger penetration depths. Our new observations add more confidence to this conceptual model.

However, the average SMO and PMO differences between the C-band and Ku-band scatterometers in this study were 10 and 12 days, respectively, which are both smaller than the 18 and 20 days of Arndt and Haas (2019), respectively. This may be related to differences between sensors, i.e. between CFOSCAT and QuikSCAT. QuikSCAT obtains backscatter coefficients from two fixed beams (46° incidence and 56° incidence) whereas we normalized the incidence angle to 40° for CFOSCAT in this
paper. Previous studies (e.g. Remund and Long, 1999; Gohin and Cavanie, 1994) have shown that the incidence angle is an important factor affecting scatterometer-based sea ice observations. The temporal resolution also affects the retrieval, possibly resulting in the smaller PMO and SMO difference between the Ku-band and C-band sensors in this study.

Our study also revealed two additional findings identical to Arndt & Haas (2019): (1) Backscatter signal exhibit latitudinal variability (Fig. 5 in this study; Fig.5 in Arndt and Haas, 2019) (2) Ku-band scatterometers are more sensitive to snowmelt (Fig. 5 of this study; Fig. 2 in Arndt and Haas, 2019).

For further comparison, we show PMO and SMO for the overlapping period of ASCAT data of these
two studies that used different data averaging methods (i.e., 2007/08 to 2015/16) in Fig. 12. For each panel of Fig. 12, the scatterplot shown on the top axis represents PMO and the bottom axis SMO. Boxplot in each panel shows the difference of results between this study and Arndt and Haas (2019). In the NWS region, the PMO from the two studies shows great consistency, while in the other three regions, our study obtained earlier PMO. This may be due to the fact that Arndt and Haas (2019) used 6-day average and
25 km resampling, smoothing out some small increase in backscatter. However, the 6-day average and 25 km resampling did not affect the detection of PMO in the NWS region due to the significant increase in backscatter coefficient amplitude in response to the snowmelt in this region (Fig. 4 and Fig. 5). As shown in the boxplot in Fig. 12e, the difference in Antarctic-wide SMO between the two studies is mostly within ±15 days, and the average value of this study is slightly later. The slightly later SMO detected in
this study is mainly due to the particularly early SMO detected by Arndt and Steffi (2019) in the SWS region. Overall, in NWS region, both PMO and SMO between these two studies are very close. Differences between the two in other regions are mainly due to differences in spatial and temporal resolution. Despite some differences, the two studies show fairly good agreement in terms of averages across all the region, demonstrating the robustness of both averaging methods.



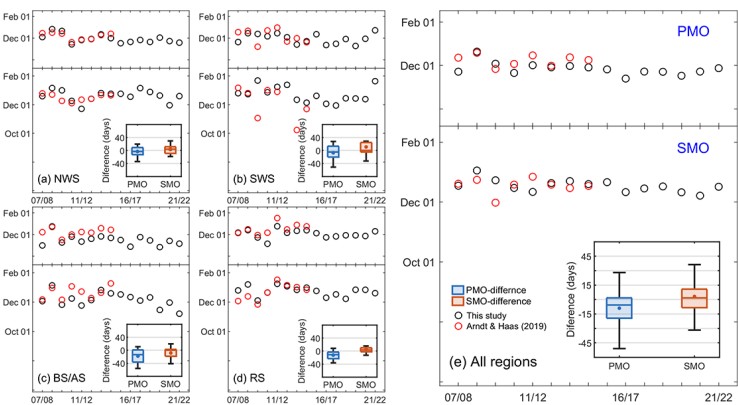


**Figure 12.** PMO and SMO derived by this study and Arndt and Haas (2019). The upper scatterplot of each panel represents the PMO results, while the lower scatterplot is the SMO results. Boxplots in each panel represent PMO and SMO differences between the two studies. Boxes are the first and third quartiles. Whiskers display the 20th and 80th percentiles. Filled circles indicate mean, and horizontal

lines median values.

### 4.3 Long-term variations of snowmelt onset dates

Having updated snow melt onset retrievals until 2021/2022 compared to the study of Arndt and Haas (2019) which ended in 2014/2015, here we also present a complete overview of long-term variations of snowmelt onset dates in our study regions. Fig. 13 shows the SMO of Arndt and Haas (2019) from

1992/1993 to 2014/2015, the ASCAT-derived SMO in this study from 2007/2008 to 2021/2022, and the CFOSCAT-derived SMO in this study from 2019/2020 to 2021/2022. The comparison of our results with those of Arndt and Haas (2019) during the overlapping period of both studies from 2007/2008 to 2014/2015 is the same as in Fig. 12 above. SMOs derived from ASCAT and CFOSCAT show consistent change patterns from 2019/2020 to 2021/2022 only in NWS and BS/AS regions. In fact, the three-year

overlap period is not sufficient to compare the consistency of SMO change patterns between the two. Therefore, we only used the ASCAT-derived SMO and SMO from Arndt and Haas (2019) to fit the long-term SMO change trend from 1992/1993 to 2021/2022. The SMO of their overlapping periods is obtained by averaging the two.



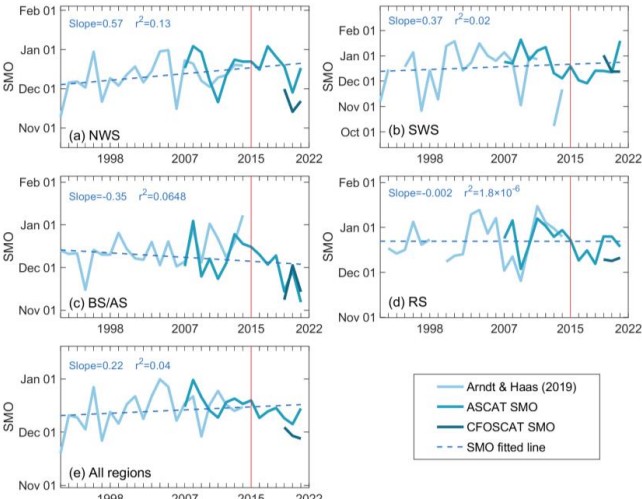

**Figure 13.** SMO time series from 1992 to 2022. Dark blue, medium blue and light blue solid lines represent CFOSCAT-derived SMO, ASCAT-derived SMO and SMO from Arndt and Haas (2019), respectively. The blue dashed line is the fitted trend line based on Arndt and Haas (2019) SMO and ASCAT-derived SMO.

From 1992/1993 to 2021/2022, SMO dates across all regions show a positive trend, i.e. occur increasingly late (Fig. 13e). However, there are large regional differences particularly in the BS/AS region where the trend is negative, i.e. SMO occurs earlier. In the RS region, there is no significant change in SMO. However, none of the shown trends are significant, as their goodness of fit $r^2$ is very small.


Interestingly mean SMO dates have remained similar after 2015 when sea ice extent began to strongly decrease around Antarctic (Fig. 13e), indicating that the sea ice retreat was not accompanied by changes in snow melt due to increased atmospheric heat or radiation fluxes. However, Fig. 13 shows strong regional variability again, and shows significantly earlier SMO dates in the Northwestern Weddell Sea

and Bellingshausen/Amundsen Seas (Figs. 13a and c). More regional studies of melt onset combined with analysis of their atmospheric and/or oceanic forcing are needed to further explore the variability of Antarctic sea ice melt.

**4.4 Snowmelt onset differences between dual-frequency scatterometers**

In this study, we applied two scatterometers with different frequencies to retrieve the snowmelt onset

dates of 12 Antarctic perennial ice study sites. Averaged across all study sites, the Ku-band scatterometer detected the snowmelt onset 10 days earlier than the C-band scatterometer. However, the difference between the snowmelt onset dates retrieved by the two scatterometers varies in different study regions. In the NWS region and RS region, the snowmelt onset of the Ku-band scatterometer is 19 and 13 days



earlier than that of the C-band scatterometer, respectively, both of which are significant, emphasizing the
idea that the Ku-band scatterometer can detect earlier snowmelt onset dates.

In the BS/AS region, due to the rapid melting in this region, the SMOs retrieved by the two scatterometers
are comparable, which has been illustrated in the Section 3.2. In the SWS region, the snowmelt onset
differences between the two scatterometers is not as significant as in the NWS, which is only 6 days.
Regions farther south experience less snowmelt attributed to the prevalent atmospheric conditions of
cold and dry air and the absence of warm air advection events from the north, suggesting there are less
melt-freeze clusters (preliminary melt stage) or superimposed ice formation farther south. The weaker
snow metamorphism and less superimposed ice formation make the increase in Ku-band backscatter
coefficient less dramatic, failing to lead to significant snowmelt onset differences between the two
scatterometers in the SWS region. Therefore, to a certain extent, the snowmelt onset difference between
dual-frequency scatterometers might be an indicator of the amount of melt-freeze clusters or
superimposed ice, i.e., the less melt-freeze clusters or superimposed ice is formed, the smaller the
snowmelt onset difference between the two scatterometers.

Additionally, we plotted the time series of SIE in February of the following year and snowmelt onset
difference between two scatterometers (hereafter referred to as SMO difference, Fig. 14) in the WS
region. The SIE is defined by areas that have a sea ice concentration of at least 15% here. In the NWS
region, the SIE in 2022 is smaller than that in 2020, while the SMO difference has a larger value in 2022
than in 2020. When the SIE is large, it might imply that there is less superimposed ice formed (Arndt et
al., 2021). This supports our speculation that the SMO difference can be an indicator of the amount of
superimposed ice. A similar phenomenon was also observed from 2021 to 2022 in the SWS region.
However, SIE is smaller in 2020 compared to 2021 in SWS region. We expected a larger SMO difference
in 2020 according to our speculation, but instead, we observed a smaller or even negative difference.
This is mainly due to the data gap in the Ku-band scatterometer data in December 2019, which resulted
in the SMO being estimated late. In the NWS region, if SIE is considered as an indicator of superimposed
ice formation, based on our speculation, the SMO difference in 2021 should be higher than observed.
Therefore, more studies of surface heat flux and in-situ observations may need to be considered to further
determine the intensity of snow metamorphism or the amount of superimposed ice to further explore the
potential of SMO differences in characterizing snow cover properties differences between NWS and
SWS regions.

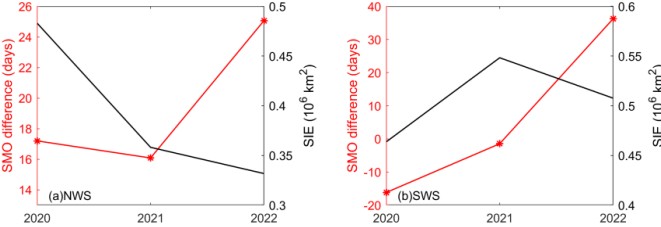

**Figure 14.** Three-year time series of snowmelt onset difference between the ASCAT and CFOSCAT,
and SIE in February of the following year in NWS and SWS regions.



**5 Conclusion**

In this study, we retrieved the pre-melt and snowmelt onset dates from 2007/2008 to 2021/2022 for 12 study regions based on backscatter data from C-band ASCAT and Ku-band CFOSCAT. Between 2019/2020 and 2021/2022, CFOSCAT detected average pre-melt and snowmelt onsets on Nov 9 and Dec 1, respectively, earlier than those of ASCAT (Nov 21 and Dec 11), which further supports the conceptual model proposed by Arndt and Haas (2019) that the Ku-band scatterometer detects the snow

metamorphism within the snowpack prior to the C-band scatterometer. However, both PMO and SMO differences between C- and Ku-band sensors from this study were smaller than those of Arndt and Haas (2019). It was mainly caused by sensor differences and different spatiotemporal resolutions. We also observed that the SMO difference between dual-frequency scatterometers is much larger in the NWS region than in the SWS region. We suggest that it is related to the difference in the intensity of snow

metamorphism or the amount of superimposed ice in these two regions. In the SWS region with weak snow metamorphism and less superimposed ice, the SMO differences are smaller. However, field measurements are needed for further verification.

         The reliability of the snowmelt detection results in this study is demonstrated by the good consistency

between the SMO of this study and that of Arndt and Haas (2019). We have also justified the used PMO and SMO identification criteria and the sea ice concentration threshold by conducting threshold and ice concentration sensitivity tests.

         Additionally, we utilized ERA5 reanalysis air temperature data and the snow buoy data to help interpret

the response of the microwave scattering signal to the snowmelt process, and to help distinguish temporal and spatial changes. Results showed that scatterometer-derived pre-melt onset was a good indicator to the initial warming and snowmelt onset was typically the start of a continuous snowmelt process due to persistent thaw-freeze cycles.

In summary, this study builds on and extends the research of Arndt and Haas (2019). Based on the SMO from 1992/1993 to 2014/2015 provided by Arndt and Haas (2019) and ASCAT-derived SMO from 2007/2008 to 2021/2022 obtained in this study, it is found that the SMO change exhibited strong interannual and regional variabilities from 1992/1993 to 2021/2022. In the context of the sharp decline in Antarctic sea ice extent in recent years, SMO did not show clear, concurrent changes, suggesting weak

relationships between the snowmelt onset dates and SIE change.

         We expect snowmelt observations from dual-frequency microwave sensors in this study will help improve the satellite-based retrieval of sea ice parameters such as sea ice thickness and sea ice type. Long-term SMO records obtained here can assist studies of polar climate change on decadal scales. Our

work also provides a more comprehensive understanding of the interaction between backscattering and the snow layer, which may aid in developing the snow models, e.g. MEMLS3&a (Microwave Emission Model of Layered Snowpacks 3&a). However, CFOSCAT stopped to provide valid data after December 2022. To provide in-depth information of long-term variability and trends in SMO and SMO differences



between dual-frequency radar observations, Ku-band scatterometer observations have to be continued
by alternative platforms, e.g., FY-3E/WindRAD, to extend the scatterometer-derived snowmelt onset
detection.

*Author contributions*. RX, CH, and CZ fomulate the overarching research goal. RX was responsible for
code implementation and data analysis. SA provided programming assistance. RX completed the paper
writing with invaluable writing guidance from CH. CH, SA, and CZ gave insightful comments.

*Competing interests*. One of the co-authors is a member in the editorial board of The Cryosphere.

*Data availability*. The CFOSCAT backscatter data were kindly provided by the National Satellite Ocean
Application Service (NSOAS, https://osdds.nsoas.org.cn/MarineDynamic). The ASCAT backscatter
data are from the Scatterometer Climate Record Pathfinder (SCP) project, sponsored by NASA
(http://www.scp.byu.edu/). Sea ice concentration data provided by University of Bremen from Oct 1,
2007 to Jun 31, 2022 were obtained from https://www.meereisportal.de (grant: REKLIM-2013-04). The
snow buoy measurements (sea ice depth, air temperature) from Jul 1, 2021 to Jan 31, 2022 were obtained
from https://www.meereisportal.de (grant: REKLIM-2013-04). The ERA5 reanalysis air temperature
data are available from the Copernicus Climate Change Service (C3S) Climate Data Store (CDS)
(https://cds.climate.copernicus.eu/cdsapp#!/dataset/reanalysis-era5-single-levels?tab=overview).

*Acknowledgements*. We acknowledge the support of the China Scholarship Council who funded a one-
year research stay of Rui Xu at the Alfred Wegener Institute in Bremerhaven, Germany. The numerical
calculations in this paper have been done on the supercomputing system in the Marine Big Data Center
of Institute for Advanced Ocean Study of Ocean University of China. The work was funded by the
National Nature Science Foundation of China, grant number 42030406, the Shandong Joint Fund for
Marine Science Research Centers, grant number U1606405 and the Marine S&T Fund of Shandong
Province for Pilot National Laboratory for Marine Science and Technology (Qingdao), grant number
No.2018SDKJ0102.

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
