# Peer review of "Dual-frequency radar observations of snowmelt processes on Antarctic perennial sea ice by CFOSCAT and ASCAT"

_EGUsphere, 2024_

## Author Comment (AC2)

**Supplementary for "Dual-frequency radar observations of snowmelt processes on Antarctic perennial sea ice by CFOSCAT and ASCAT"**

**Contents of this file**

Figures S1 to S6

Figure 5 of the manuscript

Table S1

[Figure]

**Figure S1.** ASCAT backscatter maps provided by (a) IFREMER and from (b) SIR-enhanced ASCAT product. (a) is the IFREMER ASCAT backscatter map on December 7, 2010, and (b) is the SIR-enhanced ASCAT backscatter map from December 7-December 8, 2010. The black squares represent the 12 study sites.

[Figure]

**Figure S2.** Melt onsets derived from IFREMER ASCAT data and SIR-enhanced ASCAT product. (a) shows the SMO from IFREMER ASCAT data and SIR-enhanced ASCAT for 12 study sites from 2007/2008-2021/2022, (b) shows the backscatter time series superimposed with the derived PMO and SMO for Location 1 in 2010/2011, and (c) shows the backscatter time series superimposed with the derived PMO and SMO for Location 6 in 2017/2018.

[Figure]

**Figure S3.** The time interval between the local minimum and the following local maximum when PMO and SMO are detected for 12 study sites shown by latitude during the overlapping period between ASCAT and CFOSCAT (i.e., from 2019/2020 to 2021/2022).

[Figure]

**Figure S4.** An example of interval days between the local minimum and the following local maximum when PMO and SMO were detected. The interval shown in the figure is calculated using CFOSCAT data at Location 11.

[Figure]

**Figure S5.** The ASCAT backscatter time series superimposed with the derived PMO and SMO for 9 pixels around the Location 7 in 2013/2014.

[Figure]

**Figure S6.** SMO time series from 1992 to 2021. Dark blue, medium blue and light blue solid lines represent CFOSCAT-derived SMO, ASCAT-derived SMO and SMO from Arndt and Haas (2019), respectively. The black solid line, purple dashed line, and purple dotted line are fitted trend lines for 1992-2021, 1992-2015, and 1991-2021 respectively based on the Arndt and Haas (2019) SMO and the ASCAT-derived SMO.

The following shows the Figure 5 in our manuscript:

[Figure]

**Figure 5.** Differences between seasonal maxima and minima of backscatter during the spring-summer transition period over 2019/2020 to 2021/2022 for our 12 study sites shown by latitude.

Table S1. Average time intervals (days) of ASCAT and CFOSCAT in different regions.

| Regions | ASCAT-PMO | CFOSCAT-PMO | ASCAT-SMO | CFOSCAT-SMO |
|---|---|---|---|---|
| WS | 11.6 | 12.1 | 14.0 | 12.8 |
| BS/AS | 10.1 | 8.5 | 13.6 | 12.0 |
| RS | 8.6 | 10.2 | 9.2 | 12.4 |
| All regions | 10.7 | 10.9 | 13.1 | 12.5 |

---

## Author Response (AR1)

**Response to Reviewer #1 comments:**

This well-written paper nicely confirms the well known fact that the change in snow backscatter -5 C as the temperature increases from below -5C is the result of liquid water forming on the snow crystals. As the temperature increases, more and more liquid is formed. At 0C substantial melting can occur. Heat input from this process can be from below, from the air, and from insolation.

When discussing the ASCAT products used, the authors should include a citation to the paper: R. Lindsley and D.G. Long, "Enhanced-Resolution Reconstruction of ASCAT Backscatter Measurements," IEEE Transactions on Geoscience and Remote Sensing, Vol. 54, No. 5, pp. 2589-2601, doi:10.1109/TGRS.2015.2503762, 2016.

**Response:** Thank you for your suggestion. We have incorporated the citation into the paper as you suggested. We have also included a description stating that our results confirm minimal changes in the internal properties of the snow layer at -5°C, while significant changes occur at 0°C, along with relevant citations.

**Relevant Changes Made:**

This citation appears in the text at lines 106-108 as: "Here we used gridded enhanced ASCAT data from the NASA Scatterometer Climate Record Pathfinder (SCP) project provided by Bigham Young University (BYU) that are 2-day averages with a resolution of 4.45 km×4.45 km (Long et al., 1993; Lindsley and Long., 2016)", and in the reference list at lines 735-737 as: "Lindsley, R. D., and Long, D. G. (2016).: Enhanced-resolution reconstruction of ASCAT backscatter measurements. IEEE Transactions on Geoscience and Remote Sensing, 54(5), 2589-2601. https://doi.org/10.1109/TGRS.2015.2503762, 2016."

The description stating that our results confirm minimal changes in the internal properties of the snow layer at -5°C, while significant changes occur at 0°C appears at lines 355-359 as: "This finding confirms a well-known fact that snow properties undergo minimal changes when temperatures rise above -5°C, but exhibit significant changes in their internal characteristics as temperatures exceed 0°C (e.g., Takei and Maeno, 2001; Nicolaus et al., 2009). The changes in the physical properties of the snow layer are then sensed by the scatterometer."

**Response to Reviewer #2 comments:**

**1.** Line 125. The authors used the SIR-enhanced ASCAT data with a spatial resolution of 4.45 km but resampled it onto a 12.5 km grid to match up with CFOSCAT data. Why not directly use the original 12.5 km ASCAT data (e.g. provided by IFREMER)?

**Response:** Thanks for your comments. Regarding the choice of using SIR-enhanced ASCAT data rather than other 12.5km ASCAT data such as those provide by IFREMER, this decision was primarily based on the following considerations. On the one hand, this work is an extension of Arndt & Haas (2019). To maintain consistency and allow for better comparison with the study of Arndt & Haas (2019), we used the same ASCAT

backscatter coefficient product, i.e., the SIR-enhanced ASCAT product. On the other hand, the spatial coverage of the SIR-enhanced ASCAT data is higher than that of the IFREMER ASCAT data (e.g., Fig. S1), indicating the former is expected to provide more extensive observations.

We retrieved the PMO and SMO based on IFREMER ASCAT data for 12 perennial ice study sites and compared them with those derived from SIR-enhanced ASCAT data. Figure S2a illustrates the SMO for the 12 study sites from 2007/2008 to 2021/2022 using two ASCAT data sets. It shows that the two are mostly consistent, with an absolute difference of 8 days between the two and 76% of the differences within 5 days. There are two main reasons for the difference between the two. First, there are some missing data in the IFREMER ASCAT which leads to retrieval errors of PMO or SMO (e.g., Fig. S2b). Second, in areas where snowmelt is not significant, the increase in the backscatter coefficient caused by snowmelt is inconsistent between the two data sets, leading to differences in the retrieved PMO and SMO between the two (e.g., Fig. S2c). By examining the results of all the study sites, we found that the missing IFREMER ASCAT data was the main reason for the difference between the two. In addition, missing IFREMER ASCAT data resulted in reduced retrieval rates, e.g., the SMO in Location 1 from 2014/2015-2015/2016 cannot be retrieved by IFREMER ASCAT data (Fig. S2a).

In general, even though IFREMER ASCAT data can be used directly without spatial resampling, we believe that using SIR-enhanced ASCAT data is still a better choice. This is based on two key considerations: firstly, the use of SIR-enhanced ASCAT data ensures consistency between this study and Arndt & Haas (2019); and secondly, the SIR-enhanced ASCAT data can provide more reliable observations of snowmelt onsets owing to their higher spatial coverage.

[Figure]

**Figure S1.** ASCAT backscatter maps provided by (a) IFREMER and from (b) SIR-enhanced ASCAT product. (a) is the IFREMER ASCAT backscatter map on December 7, 2010, and (b) is the SIR-enhanced ASCAT backscatter map from December 7-December 8, 2010. The black squares represent the 12 study sites.

[Figure]

**Figure S2.** Melt onsets derived from IFREMER ASCAT data and SIR-enhanced ASCAT product. (a) shows the SMO from IFREMER ASCAT data and SIR-enhanced ASCAT for 12 study sites from 2007/2008-2021/2022, (b) shows the backscatter time series superimposed with the derived PMO and SMO for Location 1 in 2010/2011, and (c) shows the backscatter time series superimposed with the derived PMO and SMO for Location 6 in 2017/2018.

**Relevant Changes Made:** We have explained the reasons for selecting the SIR-enhanced ASCAT data (referred to as SCP ASCAT data in the manuscript) in Section 2.1, lines 127-135 of the manuscript: In this study, we did not use ASCAT data at 12.5 km spatial resolution from French Research Institute for Exploitation of the Sea (IFREMER), even though employing data at the same resolution could have facilitated a more direct comparison between ASCAT and CFOSCAT. Our decision was primarily

based on the following considerations. First, this work is an extension of Arndt & Haas (2019). To maintain consistency and allow for better comparison with the study of Arndt & Haas (2019), we used the same ASCAT backscatter coefficient product, i.e., the SCP ASCAT product. In addition, there are frequent, spatial data gaps in the IFREMER ASCAT data (e.g., Fig. 2), which may result in the failure to retrieve the melt onset in some regions when utilizing this dataset. We therefore used the SCP ASCAT data instead of IFREMER data in this study.

**2.** Line 206-211. The differences between the local backscatter maximum and the preceding local backscatter minimum are the key points to identify the PMO or SMO. Are there any statistics on the interval days of minimum and following maximum for the identified PMO or SMO? Figure 2 shows that the interval may be 10-15 days for PMO and SMO. I suggest adding some statistics and discussion of the interval days. Perhaps this interval length could also represent the strength of the PMO and SMO events, apart from the amplitude of the backscatter increase.

**Response:** Thank you for your valuable feedback on our manuscript. After careful consideration and further discussion with the co-authors, we have refined our response regarding this issue. The interval length serves as a clear indicator of the progression of the melt season. To strengthen our analysis, we conducted a statistical evaluation of the interval lengths at 12 study sites and discussed the potential applications of this parameter in other facets of snowmelt process research.

Due to the drier and colder air in the south, the snowmelt intensity in the southern regions is weaker compared to the north. Therefore, we use latitude as a proxy of snowmelt intensity and give the variation of the interval length with latitude as show in Fig. S3. It can be seen that the interval lengths mostly vary from 6 days at low latitudes to 19 days at high latitudes, with a trend of increasing interval length as latitude increases. However, there are notable exceptions to this overall trend, particularly at Locations 4, 10, and 11. The particularly long interval length at Location 4 (Fig. S3b&d) is caused by the data gap of CFOSCAT data, while the short interval length at Location 10 (Fig. S3a-c) is attributed to difficulties in detecting PMO and SMO. The interval length at Location 11 also sometimes appears shorter than that at Location 12 (Fig. S3d), which is caused by the rapid change in backscatter of several pixels around Location 11 (as shown in Fig. S4, where backscatter coefficients change rapidly during occurrences of SMO). Overall, interval lengths can partially characterize snowmelt intensity, that is, stronger snowmelt intensity is associated with shorter interval lengths. However, strong snowmelt does not always equate to rapid melting. For example, when using latitude as a measure of snowmelt intensity, we observe that the melting speed at Location 11 (higher latitude) is faster than at Location 12. This highlights the complexity of the relationship between snowmelt intensity and speed.

In distinguishing between PMO and SMO, the interval length shows some potential, as indicated in Table S1, where the PMO interval lengths are consistently shorter than those of SMO. However, as illustrated in Fig. S4, the SMO interval length may sometimes be much shorter than the PMO interval length, while the backscatter rise

amplitude remains a good indicator for distinguishing between PMO and SMO.

We also found that these interval lengths do not differ much between the two scatterometers. For the entire Antarctic region, the average PMO interval lengths between ASCAT and CFOSCAT are comparable, with both around 11 days, while the average SMO interval lengths between the two differ by less than one day (Table S1). In Fig. 6 of the manuscript, we observe that differences in the amplitude of the backscatter rise between ASCAT and CFOSCAT are significant at each study sites, indicating the backscatter rise amplitude is more robust in reflecting the differences between the two scatterometers. Overall, the interval length parameter can partially indicate the intensity of snowmelt, but more validation is needed, and we believe it more directly characterizes the melt season progression or snowmelt speed. The amplitude of the backscatter rise is a more suitable detection parameter in distinguishing between PMO and SMO events and exploring the differences in the response of the two scatterometers to melt and refreezing. In the future, we may further study the snowmelt processes on Antarctic sea ice based on the interval length parameter.

[Figure]

**Figure S3.** The interval length between the local minimum and the following local maximum when PMO and SMO are detected for 12 study sites shown by latitude during the overlapping period between ASCAT and CFOSCAT (i.e., from 2019/2020 to 2021/2022).

[Figure]

**Figure S4.** An example of interval length between the local minimum and the following local maximum when PMO and SMO were detected. The interval length shown in the figure is calculated using CFOSCAT data at Location 11.

Table S1. Average time intervals (days) of ASCAT and CFOSCAT in different regions.

| Regions | ASCAT-PMO | CFOSCAT-PMO | ASCAT-SMO | CFOSCAT-SMO |
|---------|-----------|-------------|-----------|-------------|
| WS | 11.6 | 12.1 | 14.0 | 12.8 |
| BS/AS | 10.1 | 8.5 | 13.6 | 12.0 |
| RS | 8.6 | 10.2 | 9.2 | 12.4 |
| All regions | 10.7 | 10.9 | 13.1 | 12.5 |

**Relevant Changes Made:** We believe that the interval is indeed a parameter worthy of further exploration. However, the above discussion indicates that it performs less effectively than the amplitude of backscatter coefficient rise in distinguishing between PMO and SMO events, as well as in characterizing the differences between Ku- and C-band scatterometers. Therefore, we have not included all the above discussions in the manuscript. Moreover, we wish to avoid making our manuscript excessively lengthy by not incorporating all of these analyses. Instead, we briefly mentioned in Section 2.2, lines 234-240 that this parameter may help deepen our understanding of the snowmelt process: The increase in backscatter coefficient during thaw-refreeze cycles indicates the intensity of melting and refreezing, i.e. how much liquid water there has been within the snow and how much metamorphism, grain coarsening, and ice layering it has caused. Additionally, the time interval between the local minima and the subsequent local maxima when PMO or SMO occurs could be studied as potential indicators of temporal snow melt progression or rate. The time interval can provide additional context for understanding the snowmelt mechanisms on Antarctic sea ice. Although this parameter was not directly examined within the scope of this research, it bears potential significance for future inquiries.

**3.** Figure 4. I noticed some years when PMO occurred later than SMO, e.g., site 3 in Nov/Dec 2011 and site 7 in Nov/Dec 2013 and 2016, which seems abnormal. Please explain it.

**Response:** Thanks for your comment. In this study, the PMO and SMO for each study site were obtained by averaging 9 pixels around the site, as we have demonstrated in Section 2.2. At some pixels around the site, PMO can be detected but SMO cannot be detected, resulting in a PMO value being larger than the SMO value after averaging the results from 9 pixels. Figure S5 presents an example. We have added an explanation regarding this issue in the manuscript.

[Figure]

**Figure S5.** The ASCAT backscatter time series superimposed with the derived PMO and SMO for 9 pixels around the Location 7 in 2013/2014.

**Relevant Changes Made:** We explain this issue in Section 3.2, lines 275-278 of the manuscript: "It should be noted that in a few cases, Fig. 5 shows an earlier SMO than PMO, e.g., at Location 3 in 2012. This is because of the spatial variability within our 9-pixel study locations, where PMO can be detected at more pixels than SMO, which may lead to the PMO value being larger than the SMO value after averaging. An example is shown in the Appendix." We have also added Fig. S5 to the Supplementary material to further illustrate this issue.

**4.** Figure 13 and Line 522. It seems to be a declining trend of SMO for all regions after 2015. I suggest the authors add two more quantitative trend analyses for the period 1992-2015 and 2015-2022, respectively. A comparison of the two trends could add more highlights to this paper.

**Response:** Thanks a lot for your suggestion. According to your suggestion, we have fitted the trends of SMO for 1992-2015 and 2015-2021, respectively, as shown in Fig. S6.

[revised manuscript text omitted]

2415-2426, https://doi.org/10.1007/s00704-023-04820-7, 2024.

**Relevant Changes Made:** We have incorporated the above discussion into Section 4.3, lines 547-571 in the manuscript.

**Minor comments**
P: page, L: line
P4, L139. "2-m ERA5 air temperature" should be "ERA5 2-m air temperature".
P14, L373. "backscatter at ASCAT" should be "backscatter for ASCAT".
P20, L539. "differences between" should be "difference between".

**Response:** Thanks for your suggestion. We have made the necessary corrections to address the writing errors.

**Relevant Changes Made:**

In Section 2.1, lines 149-150: Here we used the **ERA5 2-m air temperature** with a spatial resolution of 0.25°×0.25°.

In Section 3.4, lines 402-403: This resulted in an increase of more than 3 dB in **backscatter for ASCAT** (Dec 26), which allowed ASCAT to detect the onset of snowmelt.

In Section 4.4, lines 581-582: In the SWS region, the snowmelt onset **difference between** the two scatterometers is not as significant as in the NWS, which is only 6 days.